# Universal Graph Continual Learning

**Thanh Duc Hoang**[*], **Viet-Tung Do**[*], **Duy-Hung Nguyen, Bao-Sinh Nguyen**

*{kris, ace, hector, simon}@cinnamon.is*

*Cinnamon AI, Vietnam*

**Huy Hoang Nguyen**                                              *huy.nguyen@oulu.fi*
*University of Oulu, Finland*

**Hung Le**                                                      *thai.le@deakin.edu.au*
*Deakin University, Australia*

**Reviewed on OpenReview:** *https://openreview.net/forum?id=wzRE5kTnl3*

## Abstract

We address catastrophic forgetting issues in graph learning as the arrival of new data from diverse task distributions often leads graph models to prioritize the current task, causing them to forget valuable insights from previous tasks. Whereas prior studies primarily tackle one setting of graph continual learning such as incremental node classification, we focus on a universal approach wherein each data point in a task can be a node or a graph, and the task varies from node to graph classification. We refer to this setting as Universal Graph Continual Learning (UGCL), which includes node-unit node classification (NUNC), graph-unit node classification (GUNC), and graph-unit graph classification (GUGC). Our novel method maintains a replay memory of nodes and neighbours to remind the model of past graph structures through distillation. Emphasizing the importance of preserving distinctive graph structures across tasks, we enforce that coarse-to-grain graph representations stay close to previous ones by minimizing our proposed global and local structure losses. We benchmark our method against various continual learning baselines in 8 real-world graph datasets and achieve significant improvement in average performance and forgetting across tasks.

## 1 Introduction

The recent literature has shown the high potential of graph neural networks (GNNs) in individual tasks from various domains, such as social and molecule networks (McCallum et al., 2000; Hamilton et al., 2017). However, when graph data come in sequence, GNNs suffer from catastrophic forgetting (French, 1999). In particular, if we treat a subgraph as a data point, a stream of data arriving from different task distributions may cause graph models to merely learn the current task using current data but forget the older ones. Thus, GNNs underperform on data from previously trained tasks. The forgetting can occur locally (node-unit) or globally (graph-unit), depending on the task, which can be node or graph classification (Galke et al., 2020; Carta et al., 2021; Wang et al., 2022).

There exist many continual learning algorithms to cope with catastrophic forgetting. Unfortunately, they are mainly designed for primitive neural networks such as multiple layer perceptron (MLP) and convolutional neural network (CNN), where the task is image classification (Kirkpatrick et al., 2017; Rusu et al., 2016). Prior works have developed methods for GNN, primarily for node incrementation (Wang et al., 2022) developed a method to cast vertex classification as a graph classification task by transforming each node of the graph into a feature graph, which might not consider relations in the original graph. Furthermore, it is unclear whether the approach can be extended to other settings involving graph incrementation. Recently, Zhang et al. (2022) have introduced a benchmark on continual learning for both node and graph classification, partly covering

---
[*]Equal contributions

incremental graph learning. However, the benchmark lacks the scenario where each data point is a graph, and the task is node classification, which can be seen in document (Park et al., 2019) and scene graphs (Tang et al., 2020).

In this work, we address continual learning in a more general context where the incoming data point can be either a node or graph and the task can be either node or graph classification. Hereinafter, we name the setting Universal Graph Continual Learning (UGCL). First, we provide a taxonomy of scenarios in UGCL consisting of node-unit node classification (NUNC), graph-unit node classification (GUNC), and graph-unit graph classification (GUGC). These scenarios have real-world significance. For instance, in NUNC, consider a social network analysis system that must continually adapt to changing user behaviors, identifying emerging trends or influencers. In GUNC, think of a document analysis system classifying text segments within evolving document graphs, essential for applications like document categorization. Lastly, in GUGC, we can have a computer vision system analyzing complex scene graphs to understand object context and relationships, crucial for autonomous driving and robotics.

We propose a new method to alleviate catastrophic forgetting in all three scenarios. Our approach adapts rehearsal and distillation techniques to graph-based models to enhance knowledge retention. We maintain a replay memory of nodes coupled with their neighbors and continuously draw samples from the memory to remind the model of previous graphs through distillation mechanisms. We argue that persevering local and global structure representations are critical for the effectiveness of graph modeling in UGCL. Hence, we enforce these representations produced by the model in a task to be close to those of the previous tasks. As such, we store models corresponding to all tasks and minimize the difference between the representations.

We demonstrate the performance of our method and other continual learning baselines in the three scenarios. For NUNC and GUGC, we employ 5 established datasets to verify our method. For GUNC, we create a new benchmark consisting of 3 public document image understanding datasets, in which a document is represented as a graph of text boxes, and the task is to extract the content and the type of these text boxes (i.e. node classification). In any case, our proposed approach outperforms the other continual learning methods significantly and shows competitive performance against the theoretical upper-bound baselines. The contributions of our study are as follows: (1) we formalize a comprehensive taxonomy of graph continual learning and propose a new method to handle all cases in the taxonomy. Our method leverages rehearsal to preserve graph data's local and global representations; (2) we create a new continual learning benchmark for graph-unit node classification, following the Class-Incremental setting with the task as node classification; (3) we conduct extensive experiments for node and graph classification to validate our method using a total of 8 datasets (see Table 6).

## 2   Related Work

Recent studies on catastrophic forgetting (CF) mainly consist of experience replay, regularization, and parameter isolation. Experience replay methods store old data via experience replay buffers and replay them while learning a new task (Isele & Cosgun, 2018). Regularization-based methods leverage additional loss terms (Febrinanto et al., 2022) to consolidate knowledge in the learning process for new tasks and keep previous knowledge. For instance, EWC (Kirkpatrick et al., 2017) keeps the network parameters close to the optimal parameters of previous tasks using L2 regularization while parameter Isolation methods (Mallya & Lazebnik, 2017; Le & Venkatesh, 2022) keep an independent model for each task to prevent any possible CF among tasks. As another example, progressive networks (Rusu et al., 2016) grow new branches for new tasks while freezing previous task parameters to avoid modification of prior knowledge. These methods are designed for simple neural networks such as MLP or CNN and are validated only in image classification. It is a non-trivial task to apply them to graph data, which are commonly found in real-world applications in a streaming fashion, such as social networks, road routes, and citation graphs.

Feature graph networks (FGNs) (Wang et al., 2022) is an early approach that addresses CF in GNNs. FGN takes the features as nodes and turns nodes into graphs so that new nodes in the regular graph turn into individual training samples. As a result, the node classification task becomes a graph classification task. Research in rehearsal methods is also applied to graph data such as ER-GNN (Zhou & Cao, 2021). This work introduced three novel approaches to selecting prior nodes to be stored in the experience buffer that

aims to prevent the forgetting phenomenon in continual node classification. ContinualGNN (Wang et al., 2020) proposed to combine both rehearsal-based methods and regularization-based methods to tackle the CF issue when the graph evolves, and new patterns appear over time. During the new pattern detection process, they store a small amount of high-influence nodes of existing patterns. Then, a regularization-based method similar to EWC is employed to keep the current model parameters do not go too far from previous model parameters so that both prior knowledge and current knowledge are maintained.

However, all the aforementioned approaches focus on one evolving graph over time. As far as our knowledge, the problem with multiple separate graphs from a task has not been studied. The closest to this task is ContinualGNN (Wang et al., 2020), formulating nodes and neighbors belonging to old patterns as one single graph. In the prior study, each data point is a single graph, but these graphs can have common nodes, whereas we aim to deal with multi-separated graphs. It is crucial to derive a method that can work in any setting, including a single big graph and multi-separated graphs. Our method can be cast as a hybrid of regularization-based and rehearsal-based methods, as it attempts to preserve knowledge inferred in previous tasks to overcome catastrophic forgetting through representation constraints and replaying. Unlike previous replay methods, however, we do not focus on which samples to replay; instead, we integrate task-agnostic local and global information from graph structures into continual learning.

## 3 Problem Formulation

In the following, we first summarize all scenarios in graph learning and their characteristics regarding continual learning. Then, we detail the problem of graph-unit continual learning, especially GUNC, which is currently underexplored in the literature.

### 3.1 Graph Learning Taxonomy

A graph is defined as $G = (V, E)$ where $V$ is the set of nodes and $E \subset V^2$ is the set of edges. Each node $v \in V$ is coupled with a feature vector $x \in X \subset \mathbb{R}^F$. Each edge $e \in E$ is associated with a weight vector $w \in \mathbb{R}^W$.

In Table 1, we categorize regular graph learning into 3 distinct tasks based on the number of graphs $N_g$ and the definition of a sample (or unit): node-unit node classification (NUNC), graph-unit graph classification (GUGC), and graph-unit node-classification (GUNC). First, in NUNC and GUNC, we learn a predictor $f$ to assign a label $y \in Y$ for every node $v \in V$, given a graph $G$ with its node features $X$ and edge weights $W$. Unlike the node-unit version working on a single big graph, our task usually involves multiple individual graphs. Big graph datasets

| Task | $N_{\mathrm{graphs}}$ | Unit | Classification |
|------|------|------|------|
| NUNC | Single | Node | Node |
| GUGC | Multiple | Graph | Graph |
| GUNC | Multiple | Graph | Node |

Table 1: The distinction of graph-unit node classification (GUNC) tacked in this work compared to node-unit node classification (NUNC) and graph-unit graph classification (GUGC).

like CORA (McCallum et al., 2000) often have a big node feature vector of size 1000 constructed from a long text paragraph which sometimes makes it less dependent on the graph topology. Meanwhile, graph-unit data extracted from datasets like CORD (Park et al., 2019) have only a smaller feature vector of size smaller than 100 made from a few words and numbers. Consequently, graph-unit learning is rather topology-dependent in this case. In a single big graph with a million nodes, we can use message-passing to propagate information from far away nodes. On the other hand, multiple graphs in graph-unit learning often have less number of nodes (about 200 nodes), thus a shallower topology. Hence, with those limitations, graph learning and continual learning with GUNC are prone to be harder than their node-unit counterparts. NUNC has been formulated and studied extensively under continual learning context (Zhou & Cao, 2021; Wang et al., 2022). The following section will focus more on GUNC and GUGC.

### 3.2 Graph-Unit Continual Learning

In the continual learning setting for GUNC and GUGC, we have the same objective as regular graph-unit learning, but the data comes in the form of a sequence $G^t = (v_i^t, e_i^t, x_i^t, w_i^t, y_i^t)_{i=1}^{M^t}$ where $t$ is the task index

and $M^t \ll M$ is the number of samples in the sequence. Assume that all items in $G^t$ belong to a distribution $P^t$ or $(v_i^t, e_i^t, x_i^t, w_i^t, y_i^t) \sim P^t$, our task is to maximize the likelihood of a label $y^t \sim P^t(y|v, e, x, w)$ . The test sample might belong to one of the past tasks or the current task $t$. Therefore, our graph-unit continual learning objective is not only to perform current task training $t$ but also not to forget past tasks $t'$ to overcome catastrophic forgetting (Li & Hoiem, 2017; Kirkpatrick et al., 2017; Rusu et al., 2016).

In particular, we focus our experiments on a specific continual learning setting called class-incremental learning (CIL). In CIL, each task contains an exclusive subset of classes $Y^t \subset Y$. According to Hsu et al. (2018), CIL is the most practical yet most difficult scenario where we have disjoint output spaces $Y^t \cap Y^{t'} = \emptyset$ between tasks which naturally leads to disjoint label distribution $P^t(Y) \cap P^{t'}(Y) = \emptyset$ and thus $P^t(G) \neq P^{t'}(G)$. We note that it differs from task-incremental learning (TIL), where the latter has a big relaxation with the known task identity or task descriptor. Therefore, one could build a multi-head architecture to treat each task separately, which totally eliminates the risk of cross-task prediction.

## 4 Method

In this work, we aim to develop a system to universally overcome catastrophic forgetting for NUNC, GUGC, and GUNC. The workflow of the proposed system is shown in Figure 1. Given an input graph and embedded feature of nodes, our experience replay (**ER**) backbone selects replay samples and stores them in a **Replay Buffer** (Rolnick et al., 2018). Then, we utilize Local Structure Distillation and Global Structure Distillation to remind the model of the global-local structure of graph samples in previous tasks. The backbone of our method is based on the rehearsal approach, which often dominates in continual learning because the replayed samples provide direct information about past tasks, giving huge advantages against regularization-based approaches. Whereas most prior replay mechanisms focus on which samples to replay, we investigate another aspect. As such, we argue that aside from task-related information, additional knowledge can be derived from replayed samples in the form of a task-agnostic distillation loss. Although we discuss only one replay mechanism as the backbone, our distillation mechanism is not limited to any specific one.

### 4.1 Graph Experience Replay

Experience Replay (ER) is well-known in the continual image classification setting with its simple implementation yet effectiveness. Regarding graph learning, the ER adaptation requires more or fewer changes depending on the specific learning task. Similar to the image domain, ER for graphs (Zhou & Cao, 2021) stores a portion of data belonging to classes from past tasks in the replay buffer $G_b$. Then, in each training step of the current task, we query $k$ samples to optimize together with the original training data to prevent catastrophic forgetting. For GUGC, we store graphs belonging to classes from past tasks with all of their nodes in the replay buffer. For GUNC, although we only need nodes belonging to classes from past tasks, we also store their corresponding graphs to use the connectivity information for later computation. Hence one item (either node or graph) in our replay buffer always comes with an associated graph. For NUNC, we apply a similar procedure except that all nodes are from a single graph. Therefore, unlike NUNC having all nodes connected to each other, GUNC has only some connections from nodes in the same graph, thus making it harder to learn (see more in Sec 3). In the following, we describe the implementation of the replay buffer.

First, our replay buffer $G_b$ is implemented as a fixed-size list whose each element is a graph/ node with its label $y \in Y^t$. To expand $G_b$, we collect all training data in $G^t$ after finishing the training on $G^t$. At this point, our replay buffer $G_b$ might become bigger than the allowed size $\delta$, which triggers a replacement procedure.

The replacement strategy iterates between 2 steps: (1) finding the class that has the biggest number of samples in the replay buffer, and (2) removing one sample belonging to that class to ensure class balance (Prabhu et al., 2020). To choose which node to remove, we can use a random selection procedure or a ranking selection like Mean Feature and Coverage Maximization (Zhou & Cao, 2021). Finally, with the replay buffer $G_b$, we can optimize our model continually, minimizing the total loss $\mathcal{L}^{Total}$, which is a convex combination of the current downstream task loss $\mathcal{L}$ and the continual learning loss $\mathcal{L}^{CL}$ focusing on previous tasks as follows,

$$\mathcal{L}^{Total} = \alpha\mathcal{L} + (1-\alpha)\mathcal{L}^{CL} \tag{1}$$

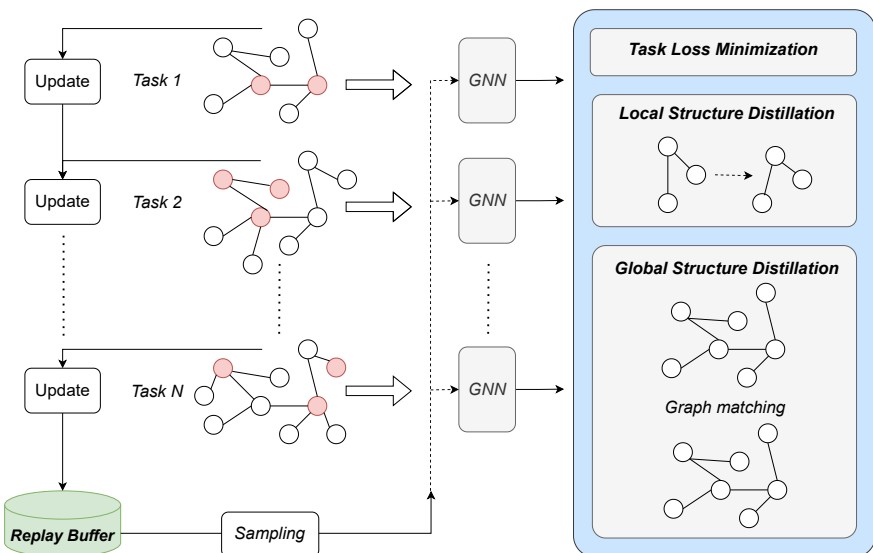

Figure 1: Overview of the proposed method. For simplicity, we show $N$ tasks, where the model will be continuously learned and updated. The left side gives the building and updating of the Replay Buffer (RB) across tasks. While the right side describes the replay and knowledge distillation process. Nodes/Graphs sampled from RB, along with those in the current task, will be used to gain structured knowledge from the previous task model by using Local Structure Distillation and Global Structure Distillation processes. The RB will be expanded with current task graphs by replacement strategies.

Traditionally, $\mathcal{L} = L(f(v, e, x, w), y)$ s.t. $(v, e, x, w, y) \sim G^t$, and $\mathcal{L}^{CL} = \mathcal{L}^{ER} = L(f(v_b, e_b, x_b, w_b), y_b)$ s.t. $(v_b, e_b, x_b, w_b, y_b) \sim G_b$, where $\mathcal{L}^{ER}$ is known as experience replay loss, $f$ is the predictor, and $L$ is the conventional cross-entropy. We use a uniform sampling strategy to sample examples from the replay buffer $G_b$ for simplicity. Here, $\alpha$ is the coefficient balancing the learning of the current and past tasks, which can be tuned during training. However, to reduce the hyperparameter tuning cost, we compute it as $\alpha = |G_b|/(|G_b| + |G^t|)$. In the next sections, we propose to enhance the continual learning loss $\mathcal{L}^{CL}$ with our new local and global structure losses.

## 4.2 Local-Global Knowledge Distillation

As graph learning is mainly about iterative feature aggregation or message passing across local neighbors, transferring learned graph structure representations is critical to retain a graph model performance. Since the experiment replay strategy reminds the model of previous task samples within the task objective, it may not help explicitly transfer graph topology structure representation learned by trained models. Inspired by distillation approaches (Hinton et al., 2014; Yim et al., 2017), we propose a way to effectively transfer the acquired structure knowledge from models trained in old tasks. Prior distillation strategies extract the output (Hinton et al., 2014), intermediate activation (Zagoruyko & Komodakis, 2016; Huang & Wang, 2017), or label (Hou et al., 2019; Douillard et al., 2020) from CNN models with image inputs. Unlike them, our method focuses on graph neural networks, which handle a more general input from mixed domains (visual, text, molecules). While there are some studies (Deng & Zhang, 2021; Yang et al., 2020) try to distill knowledge from previous graph models to a new model, they merely focus on learning the structure of the input graph or only learning local structure. Our work aims to transfer global and local characteristics of graph structure representations embedded in trained models to a model in the new task, thereby preventing the model from forgetting key characteristics of past tasks.

### 4.2.1 Local Structure Distillation

Aggregating neighbor information plays an important role in graph learning, which depends much on typical local context to make the prediction. Therefore keeping the model from forgetting previously learned local context or structure between a node and its neighbors is necessary. We hypothesize that the model trained on previous tasks has already encoded the local structure representations of past tasks. Hence, using past samples from the replay buffer can reconstruct the representations in the past models. These representations will be used to enforce the current model to output similar local structure representations as the past models'.

Let $g$ denote the graph associated with the item sampled from the replay buffer. We use this graph to extract the neighbors of the considering node to build the local structure representations. We construct a representation of the local structure through 2 steps: (1) extracting node representations and (2) forming local structure representations.

Given a set of node features $X = \{x_1, x_2, \ldots, x_N\}$ from the graph $g$ containing $N$ nodes and an adjacency matrix $A$ that represents the edges among nodes. The hidden representation of node $v_i$ at the $l$-th layer, denoted as $\mathbf{z}_i^{(l)}$, is formulated as $\mathbf{z}_i^{(l)} = \sigma(\sum_{j \in K_{(i)}} A_{ij} \mathbf{z}_j^{l-1} W^l)$ where $K(i)$ denotes the neighbors of node $v_i$, $\sigma(\cdot)$ is an activation function, and $W^l$ is the transformation matrix of the $l$-th layer. $\mathbf{z}_i^{(0)}$ represents the input feature of node $v_i$. $W^l$ is a weight matrix that defines the aggregation strategy from neighbors. Inspired by Mcpherson et al. (2001), we represent the local structure representation by measuring the difference between the features of a node and the features of neighbor nodes. More specifically, the local structure difference would be formulated as follows

$$S_{(g,i)}^c = \mathbf{z}_i^c - \frac{1}{N_i} \sum_{n=1}^{N_i} \mathbf{z}_n^c \tag{2}$$

$$S_{(g,i)}^o = \mathbf{z}_i^o - \frac{1}{M_i} \sum_{m=1}^{M_i} \mathbf{z}_m^o \tag{3}$$

Eqs. (2) and (3) build the local structure embedding vectors as the difference between a node feature and its neighbor mean feature, where $\mathbf{z}_i^c$ and $\mathbf{z}_i^o$ are node features from the current model and the previous model, respectively. $\mathbf{z}_n^c$ and $\mathbf{z}_m^o$ are neighbor node features from the current and old models, respectively. $N_i$ and $M_i$ are the neighbor size where neighbors were selected based on their connection with the node $v_i$ which is defined in the adjacency matrix $A$. We then try to minimize the distance between two embeddings by minimizing the following Local Structure Loss

$$\mathcal{L}^{LS} = \frac{1}{K_g \cdot K_n} \sum_{j=1}^{K_g} \sum_{i=1}^{K_n} \left[ 1 - \cos\left( S_{(j,i)}^c, S_{(j,i)}^o \right) \right] \tag{4}$$

where $cos$ is the cosine similarity, $K_g$ is the number of associated graphs sampled from the replay buffer, and $K_n$ is the number of nodes sampled from those graphs. This mechanism prevents the current model from forgetting previously learned local structure representation in previous tasks while not interfering directly with the learning of the current task since the regularization occurs only in feature space, not in the final prediction layer, as in Experience Replay. The choice of $K_n$ and its impact will be discussed in the Experiment section.

### 4.2.2 Global Structure Distillation

The latent interactions among non-neighbor nodes also play an essential role in learning tasks that require global interactions, such as graph classification. Preserving the global structure of how the previous model embeds would keep the current model from forgetting such information learned in previous tasks. We explicitly train the current model to preserve the global structure to distil representational knowledge from the previous model better. To achieve this, we introduce the Global Structure Loss to maximize similarities among all node features.

Specifically, we distil the representations learned from the previous task model by computing similarities between graph embeddings from the current and the previous task's model as follows

$$\mathcal{L}^{GS} = \frac{1}{K_g} \sum_{j=1}^{K_g} \left[ 1 - \cos\left(f_j^P, f_j^C\right) \right] \tag{5}$$

where $f_j^P$ and $f_j^C$ are the graph embeddings from the previous and the current task's model, respectively, where they are computed by applying weighted sum and max pooling to the node features generated by the models followed by a linear layer. $K_g$ is the number of graphs retrieved from the replay buffer, similar to that in the local structure distillation. Here, replaying graph samples of previous tasks prevents the current model from forgetting learned global structure information in previous tasks, which assists the representation learning of the entire graph distribution.

Finally, the continual learning loss $\mathcal{L}^{CL}$ becomes a linear combination of the local, global structure and the experience replay loss:

$$\mathcal{L}^{CL} = \mathcal{L}^{ER} + \beta \mathcal{L}^{LS} + \gamma \mathcal{L}^{GS} \tag{6}$$

where $\beta$ and $\gamma$ are hyperparameters to be tuned while training.

## 5 Experiments

### 5.1 Implementation Details

**Model Architecture**  Both node classification (GUNC, NUNC) and graph classification (GUGC) share the same graph model architecture as a simple graph convolutional network (GCN) with 3 GCN layers (Kipf & Welling, 2016). Before being fed to the first GCN layer, the input features go through a linear layer to learn a more compact representation. Except for the first layer, each GCN layer has a residual connection from its input to its output. For GUGC benchmarks, we add a WeightedSumAndMax (Zhang et al., 2022) pooling layer to calculate the graph embedding. Finally, we pass the final GCN output through a linear layer with a softmax activation to output the multiclass probability for each node/graph.

**Training Protocol**  We follow the traditional setting of Class Incremental learning. Specifically, we first train the GNN with either node classification or graph classification objective on the first task consisting of $N_{\text{classes}}^1$ classes. Next, from the second task to the last task, we keep expanding the model output layer with additional $N_{\text{classes}}^t$ units. Strictly, we only provide the current task labels, even though our model has past task output units. As a result, the model performance vastly degrades if no mechanism is implemented to prevent catastrophic forgetting. Additionally, we keep all the model checkpoints after every task to run the evaluation later. Besides, keeping past checkpoints is necessary for our distillation strategies.

**Evaluation Protocol**  After training our model on each task, we evaluate its performance on the testing data of the current task and all previous tasks. To be in line with Chaudhry et al. (2018); Lopez-Paz & Ranzato (2017); Zhang et al. (2022), we also utilize the metrics Average Performance (AP) and Average Forgetting (AF).

**Baselines**  We extensively compare our method with non-continual learning baselines and continual learning baselines. **Non-CL baselines:**

- **Finetune** (Girshick et al., 2013): This baseline follows the above training and evaluating protocols without any further mechanisms to prevent catastrophic forgetting. This baseline is a basic starting point for further integration of advanced continual learning techniques.

- **Feature extraction** (Donahue et al., 2013; Razavian et al., 2014): The baseline trains the model on the first task dataset and then freezes all previous layers while adding new output units to the classification layer (see Training Protocol). With this baseline, we want to validate whether a model can still learn a new task without forgetting past tasks by keeping the same representation.

- **Independent** (Paz & Ranzato, 2017): Instead of expanding the model, we train a whole new GNN with $N_{\text{classes}}/N_{\text{tasks}}$ output units for every encountering task. As a result, the Forgetting Measure for this baseline is always zero. At the same time, the Average Accuracy is calculated by averaging the current task accuracies but not the final ones like the other baselines:

$$A_{N_{\text{tasks}}} = \frac{1}{N_{\text{tasks}}} \sum_{j=1}^{N_{\text{tasks}}} a_{j,j}$$

  Often, the negative impact of forgetting is bigger than the positive impact of the forward transfer. Thus, the result of this baseline is considered an upper bound for other methods.

- **Joint training** (Caruana, 1998): This baseline is very similar to Finetune, but we keep the past datasets alongside the current task dataset to train the model. Therefore, the result of this baseline is also another upper bound for other methods.

**CL baselines**. By adding different mechanisms to prevent catastrophic forgetting over the Finetune baseline, we obtain several advanced baselines: LwF (Li & Hoiem, 2017), DK (Hinton et al., 2015), EWC (Kirkpatrick et al., 2017), MAS (Aljundi et al., 2018), TWP (Liu et al., 2021), GEM (Lopez-Paz & Ranzato, 2017), and ER (Rolnick et al., 2018). Since some of the methods are originally for non-graph data, we slightly adapt them to our tasks as follows:

- In **LwF**, we strictly follow the original implementation with the distillation loss calculated based on the output of the current model and the model in the previous task for an input graph in our current training data. Specifically, given an input graph, we calculate the distillation loss for all node features.

- **DK** is implemented similarly, except for the loss function changes to KL Divergence Loss (Joyce, 2011).

- For **EWC**, we also follow the original implementation that employs the Fisher information matrix to regularize the loss function so that network parameters are close to the learned parameters of previous tasks.

- **MAS** can be implemented similarly to **EWC**, except we use the gradient of the output with respect to a parameter to evaluate its importance.

- On the other hand, **TWP** is designed for graphs with a loss to preserve the topological information of the graphs between tasks.

- For **GEM**, we store training samples (graphs/ nodes) in the replay buffer to then modify the current task gradients with the gradients calculated with the replay buffer data.

- **ER-GNN** is implemented in (Zhou & Cao, 2021). This method is the direct adaptation of the well-known Experience Replay mechanism in classical continual learning to the new domain- graphs.

- **ER** is the base method for our work that was already described in Sec 4.1 which is highly inspired by **ER-GNN** but with the additional replacement procedure. For the replacement strategy, we use random selection. We note that since GUGC and NUNC are common settings in the literature (Zhang et al., 2022), we can implement or reuse baselines on those settings than on the less well-known setting GUNC.

**Experience Replay with Structure Distillation (Ours)**. Our method integrates task-agnostic knowledge into ER. Like other ER-related methods, the replay buffer size acts as one hyperparameter. Moreover, to apply Structure Distillation to NUNC, GUNC, and GUGC, we loop through a set of $T_g$ graph samples from both the replay buffer and input graphs. In our experiments, we sampled graphs from the replay buffer equal to the batch size of the input graph for distilling structure knowledge. Similar to other baselines like LwF and DK, the distillation loss in our method transfers naturally from node classification to graph classification. In total, we examine 3 variants, which are ER with global structure distillation (**ER-GS**), ER with local structure distillation (**ER-LS**), and ER with both global-local structure distillation (**ER-GS-LS**).

| Method | CORD | | SROIE | | Wild Receipt | |
|---|---|---|---|---|---|---|
| | AP (%) ↑ | AF (%) ↑ | AP (%) ↑ | AF (%) ↑ | AP (%) ↑ | AF (%) ↑ |
| Finetune | $15.3 \pm 0.3$ | $-76.7 \pm 0.2$ | $12.4 \pm 0.4$ | $-80.1 \pm 0.1$ | $23.6 \pm 0.1$ | $-78.8 \pm 0.1$ |
| Feature Extraction | $19.6 \pm 1.3$ | $-8.3 \pm 1.7$ | $22.9 \pm 0.2$ | $-11.9 \pm 0.8$ | $23.6 \pm 0.1$ | $-10.5 \pm 0.6$ |
| Independent | $79.9 \pm 1.5$ | - | $84.9 \pm 0.6$ | - | $84.6 \pm 0.1$ | - |
| Joint | $90.9 \pm 0.8$ | - | $80.9 \pm 1.4$ | - | $83.1 \pm 0.3$ | - |
| EWC | $14.5 \pm 2.0$ | $-33.0 \pm 1.6$ | $12.6 \pm 0.2$ | $-35.4 \pm 0.1$ | $19.3 \pm 0.4$ | $-34.1 \pm 1.3$ |
| LwF | $15.5 \pm 1.0$ | $-16.0 \pm 0.9$ | $13.8 \pm 1.0$ | $-32.9 \pm 1.7$ | $20.2 \pm 0.5$ | $-45.8 \pm 8.7$ |
| DK | $15.9 \pm 0.3$ | $-59.9 \pm 2.2$ | $11.1 \pm 0.2$ | $-67.8 \pm 5.6$ | $22.6 \pm 0.4$ | $-76.8 \pm 0.1$ |
| ER (Ours) | $79.4 \pm 0.4$ | $\underline{-2.3 \pm 0.3}$ | $81.0 \pm 1.4$ | $-1.5 \pm 0.2$ | $81.6 \pm 0.3$ | $-2.4 \pm 0.4$ |
| ER-GS (Ours) | $\underline{82.7 \pm 1.1}$ | $-2.9 \pm 0.5$ | $84.0 \pm 0.7$ | $\underline{-1.2 \pm 0.2}$ | $\underline{82.4 \pm 0.1}$ | $\underline{-2.0 \pm 0.2}$ |
| ER-LS (Ours) | $80.4 \pm 1.1$ | $-3.6 \pm 0.3$ | $\underline{84.2 \pm 0.2}$ | $-1.3 \pm 0.3$ | $82.3 \pm 0.3$ | $-2.3 \pm 0.4$ |
| ER-GS-LS (Ours) | $\mathbf{84.0 \pm 0.4}$ | $\mathbf{-1.9 \pm 0.3}$ | $\mathbf{84.4 \pm 0.3}$ | $\mathbf{-1.2 \pm 0.2}$ | $\mathbf{83.0 \pm 0.1}$ | $\mathbf{-1.2 \pm 0.6}$ |

Table 2: Performance comparisons on the task of Document Image Understanding. **Bold**/underlined denote the best/second best-performing CL technique for each column. We note that non-CL baselines (first rows) serve as theoretical bounds for reference purposes. $-$ denotes ignored AF calculations.

## 5.2 GUNC: Document Image Understanding

**Datasets**    We consider 3 document image understanding datasets: SROIE (Huang et al., 2019), CORD (Park et al., 2019), and WILDRECEIPT (Sun et al., 2021) (Appendix Table 6). We can set up graph-unit node classification on document datasets with a few heuristic steps (refer to Task Description). As mentioned, document image understanding datasets often have less informative feature vectors and shallow topology, which causes difficulty for graph-unit node classification. For example, in CORD, the maximum number of nodes is 84, and the input feature vector size is 51. For SROIE and WILDRECEIPT, the quantities are $(240, 43)$ and $(217, 81)$ respectively. Thus, doing continual learning for GUNC for those document datasets is rather challenging.

In the graph-unit node classification task, given the input nodes with associated features and also the graph, the task is to generate the embedded feature for each node such that nodes with different classes can be separated. We adopt CORD, SROIE, and WILDRECEIPT datasets that contain, 30, 4, and 12 classes, respectively. For CORD, we keep the original train-validation-test split while we further split the training data of SROIE and WILDRECEIPT with a ratio 80-20 for the training and validation set.

**Task Description**    For document image understanding using graph-unit node classification (GUNC), we first build a graph where each node is equivalent to a text box. Subsequently, the label for each node is the category of the corresponding text line. For example, in CORD (Park et al., 2019), we have categories like store_name, price. Next, to form the edge information and build an adjacency matrix, we adopt a heuristic rule based on the horizontal/vertical alignments between nodes as well as their distances (Qian et al., 2018). Particularly, since different documents have different sizes, we use a relative distance compared with the document image size to justify an edge.

After that, with the set of distinct classes as $C$, we construct $N_{\text{tasks}}$ disjoint same-length subsets $(G^t)$ as $N_{\text{tasks}}$ continual learning tasks. In each subset, we ignore nodes with labels $y \notin C^t$ in the training loss for that set. A detailed task setting is shown in Appendix Table 6 for GUNC, including SROIE, CORD, and WILDRECEIPT. Often, in GUNC, we leverage non-target nodes as negative samples to boost the overall accuracy. Therefore, in GUNC, tasks can have only 1 class like in the case of SROIE without losing generality.

**Hyperparameter Tuning**    We use grid-search to tune common hyperparameters. Particularly, the number of epochs is sampled in the range of $[50, 100]$ with a discretization step of 10, and the learning rate of Adam optimizer is sampled from the set $\{10^{-2}, 10^{-3}, 10^{-4}, 10^{-5}\}$. For EWC, LwF, and DK, the old-new task weight is sampled in the range of $[0.1, 0.9]$. For ER, we set the replay buffer size to 1000 nodes from all classes. For our methods, we optimize the loss weights $\beta$ and $\gamma$ in the range of $[0.1, 0.9]$ using a faster hyperparameter search: Bayesian Optimization (Wu et al., 2019) and realize that balance weights $\beta = \gamma = 0.5$ work best. For

| Method | ENZYMES | | Aromaticity | |
|---|---|---|---|---|
| | AP (%) ↑ | AF (%) ↑ | AP (%) ↑ | AF (%) ↑ |
| Finetune | $23.0 \pm 0.5$ | $-47.5 \pm 7.5$ | $5.4 \pm 0.2$ | $-66.3 \pm 2.0$ |
| Joint | $93.2 \pm 6.2$ | - | $77.2 \pm 1.3$ | - |
| EWC | $17.2 \pm 0.9$ | $-51.1 \pm 6.1$ | $7.5 \pm 1.6$ | $-69.9 \pm 2.5$ |
| MAS | $21.6 \pm 1.6$ | $-20.0 \pm 2.5$ | $7.8 \pm 1.0$ | $-80.5 \pm 2.1$ |
| GEM | $20.0 \pm 0.0$ | $-57.5 \pm 1.0$ | $9.3 \pm 2.4$ | $-75.6 \pm 1.1$ |
| TWP | $22.5 \pm 0.9$ | $-56.5 \pm 6.5$ | $6.5 \pm 1.6$ | $-45.4 \pm 7.7$ |
| LwF | $21.0 \pm 1.1$ | $-55.1 \pm 5.2$ | $5.7 \pm 3.3$ | $-16.1 \pm 4.7$ |
| ER-GNN | $21.3 \pm 1.3$ | $-42.5 \pm 4.5$ | $32.5 \pm 1.7$ | $-18.2 \pm 1.6$ |
| ER (Ours) | $22.5 \pm 0.9$ | $-35.0 \pm 2.5$ | $37.2 \pm 1.5$ | $14.0 \pm 1.3$ |
| ER-GS (Ours) | $21.7 \pm 2.1$ | $-35.0 \pm 7.2$ | $35.4 \pm 1.6$ | $6.2 \pm 0.7$ |
| ER-LS (Ours) | $\underline{23.3 \pm 1.7}$ | $\underline{-25.0 \pm 7.5}$ | $\underline{40.0 \pm 1.7}$ | $\underline{14.5 \pm 0.6}$ |
| ER-GS-LS (Ours) | $\mathbf{25.5 \pm 1.7}$ | $\mathbf{-20.0 \pm 5.1}$ | $\mathbf{45.4 \pm 1.5}$ | $\mathbf{17.6 \pm 1.2}$ |

Table 3: Performance comparisons under task Graph Classification on different datasets. **Bold**/underlined denote the best/second best-performing CL technique for each column. We note that non-CL baselines (first rows) serve as theoretical bounds for reference purposes. $-$ indicates ignored AF calculation.

ER-LS, and ER-GS-LS, we set $K_n$ equal to batch size which is one as graphs in the document are not large and dense compared to other datasets.

**Results** The results on the 3 document image understanding datasets CORD, SROIE, and WILDRECEIPT are shown in Table 2. Through reusing past samples from previous tasks, ER can largely improve the performance of the fine-tuning strategy and is significantly better than baselines, including EWC, LwF, DK, and Feature Extraction. Particularly, the model can remember a past task by utilizing a set of past representative samples in the replay buffer. This direct strategy turns out to be more effective with graph data since graph-unit continual learning often contains distinct tasks, which is unlike image classification, where different tasks share similar structures like digits (MNIST (LeCun et al., 1998)). For instance, a task in CORD can contain address information while another task can contain price. As a result, regularization-based methods like EWC, LwF, and DK tend to perform poorly on our GUNC benchmark. Our method further improves the vanilla ER with the ability to transfer the global and local structure information, resulting in the best performance among all the comparison methods.

Regarding evaluation metrics, the average improvements of our method over Finetune on CORD, SROIE, and WILDRECEIPT are 68.7%, 72.0%, and 59.4%. As stated in the baseline subsection, joint training can be considered an ER-based method whose replay buffer contains all past data. Thus, it is reasonable that our proposed method, in general, has lower performance than this upper bound baseline. Surprisingly, SROIE benefits most from our framework, which performs even better than joint training. However, without the local-structure (LS) and global-structure (GS) distillation, ER baseline alone only has a similar performance as the joint training on SROIE. That indicates the importance of introducing additional topology information in the form of LS and GS. Besides, our proposed method requires fewer samples than the Joint baseline, which results in a faster training speed. Formally, a joint training uses $\frac{N_{\text{tasks}}(N_{\text{tasks}}+1)}{2}$ task data while our method uses only $N_{\text{tasks}}$ task data. Thus, our method is $\frac{(N_{\text{tasks}}+1)}{2}$ times the speed of the joint training baseline. For example, in CORD, $N_{\text{tasks}} = 5$, we have a 3 times faster training speed. Note that we ignore the cost for the replay buffer because its fixed size does not scale with the number of tasks.

The improvements mainly come from incorporating information from both the data and graph learned structure representation, which provides additional supervision that can fully utilize the graph structures and limited samples. Specifically, our methods are remarkably less forgetting compared to other baselines with AF of $-1.9\%$, $-1.2\%$, and $-1.2\%$ in CORD, SROIE, and Wild Receipt, respectively. The results are also nearly approach results of Independent and Joint approaches that require all training samples across tasks and more computing resources.

## 5.3 GUGC: Molecule Classification

**Datasets** We consider 2 molecule graph classification datasets: ENZYMES, and Aromaticity (Xiong et al., 2019) (see details in Appendix Table 6). ENZYMES is a dataset of protein tertiary structures which contains

600 enzymes. The task is to classify each enzyme into one of the 6 Enzyme Commission numbers (EC number). Aromaticity (Xiong et al., 2019) is a molecule dataset with the task of predicting the number of aromatic atoms in molecules. The node features and adjacency matrices are available in the dataset. We split the data into training/validation/test sets with a ratio of 80/10/10.

**Task Description**  For graph classification using graph-unit graph classification (GUGC), we are given the processed graphs with all the necessary information. Besides, unlike GUNC, we do not have to filter out non-target nodes of each task since the labels of each graph group each subset. As there is no None/other class in graph labels, we have to ensure at least 2 classes are presented in each task, so we just group 2 classes into one task incrementally, resulting in 3 and 15 binary classification tasks for ENZYMES and Aromaticity, respectively.

**Hyperparameter Tuning**  To tune the hyperparameters, we follow exactly the procedure in (Zhang et al., 2022) for EWC, MAS, GEM, TWP, and LwF. For ER, we use 10 graphs as the buffer size for each class in small datasets like ENZYMES, while big datasets like Aromaticity use 100 graphs. For our methods, based on our experience with the previous task, we set the loss weights $\beta$ and $\gamma$ as 0.5 and 0.5, respectively. In ER-LS and ER-GS-LS, we set $K_n = 15$ as the graph size in this task is much bigger compared to the graph in the GUNC task.

**Results**  We report the results on 2 datasets ENZYMES and Aromaticity in Table 3. We strictly follow the class-incremental setting, training procedure, and data processing introduced in a GUGC benchmark (Zhang et al., 2022). Overall, the performance of Finetune method is among the worst both in terms of AP and AF. While our ER variants still demonstrate outstanding performance. Specifically, on the ENZYMES dataset, we observe a big gap of 27.5% AF and a slight improvement of 2.5% AP compared to Finetune. On Aromaticity, big improvements happen in AP (40.0%) and AF (83.9%). Furthermore, the positive AF shows that our method can overcome not only catastrophic forgetting but also bring positive forward transfer. Finally, ER-GS-LS achieves the best performance compared to other ER-based methods. This proves the usefulness of our Local-Global Knowledge Distillation, even in the graph-level classification task.

### 5.4  NUNC: Social Graph Learning

**Datasets**  We consider 3 public datasets: CoraFull (McCallum et al., 2000), Reddit (Hamilton et al., 2017), and Arxiv[1]. We strictly follow the class-incremental setting, training procedure, and data processing introduced in a NUNC benchmark (Zhang et al., 2022).

**Hyperparameter Tuning**  To tune the hyperparameters, we follow strictly the procedure in Zhang et al. (2022) for EWC, MAS, GEM, TWP, and LwF. For ER, we use 10 nodes as the buffer size for each class in small datasets like CoraFull, while big datasets like Reddit use 100 nodes. For our methods, based on our experience with the previous tasks, we set the loss weights $\beta$ and $\gamma$ as 0.5 and 0.5, respectively. In LS related method, we set $K_n = 15$ for CoraFull and Arxiv dataset, while on Reddit we set $K_n = 5$. We provide details of performance for some choices of $K_n$ in Table 5.

**Results**  As shown in Table 4, we compare our methods with other baselines in the benchmark. Overall, on both Arxiv and Reddit datasets, the performance of our proposed methods including ER-GS, ER-LS, and ER-GS-LS surpasses other advanced baselines like EWC, MAS, or ER by a large margin. On Corafull, our ER-GS-LS ranks highest with a substantial gap among CL baselines followed by TWP, we create a significant gap of 36.6% AP and 39.6% AF to the Finetune baseline. On top of that, our method also gain the highest scores on the rest two datasets, Arxiv and Reddit. Specifically, on Arxiv, the gap is 29.9% AP and 63.1% AF to the Finetune baseline, and our method can come even close to the upper bound baseline Joint with a gap of 5.9% AP. For Reddit, our Local Structure Distillation (ER-LS) is suboptimal due to the densely connected nodes in a graph, which makes global information to be more favored in this case. This has been proven to be truly effective in the Global Structure Distillation setting. Then combining local structure information can

---

[1]https://ogb.stanford.edu/docs/nodeprop/#ogbn-arxiv

| Method | CoraFull | | Arxiv | | Reddit | |
|---|---|---|---|---|---|---|
| | AP (%) ↑ | AF (%) ↑ | AP (%) ↑ | AF (%) ↑ | AP (%) ↑ | AF (%) ↑ |
| Finetune | $2.9 \pm 0.0$ | $-94.7 \pm 0.1$ | $4.9 \pm 0.0$ | $-87.0 \pm 1.5$ | $5.1 \pm 0.3$ | $-94.5 \pm 2.5$ |
| Joint | $80.6 \pm 1.5$ | - | $40.7 \pm 2.3$ | - | $84.3 \pm 1.7$ | - |
| EWC | $3.7 \pm 0.6$ | $-93.0 \pm 1.2$ | $4.9 \pm 0.0$ | $-88.9 \pm 0.3$ | $10.6 \pm 1.5$ | $-92.9 \pm 1.6$ |
| MAS | $2.9 \pm 1.2$ | $-88.3 \pm 2.1$ | $4.8 \pm 1.1$ | $-82.8 \pm 2.4$ | $9.7 \pm 1.3$ | $-93.7 \pm 2.2$ |
| GEM | $6.4 \pm 1.5$ | $-90.0 \pm 2.1$ | $4.9 \pm 1.2$ | $-86.4 \pm 2.4$ | $28.4 \pm 3.5$ | $-71.9 \pm 4.2$ |
| TWP | $13.1 \pm 2.1$ | $-80.7 \pm 2.3$ | $4.8 \pm 0.9$ | $-88.3 \pm 1.5$ | $9.3 \pm 1.2$ | $-91.3 \pm 2.3$ |
| LwF | $2.7 \pm 0.7$ | $-93.7 \pm 1.6$ | $4.9 \pm 1.1$ | $-87.0 \pm 2.3$ | $5.2 \pm 1.5$ | $-81.0 \pm 1.8$ |
| ER-GNN | $2.1 \pm 1.1$ | $-91.6 \pm 1.7$ | $26.8 \pm 1.4$ | $-52.4 \pm 2.1$ | $71.5 \pm 5.2$ | $-28.6 \pm 5.2$ |
| ER (Ours) | $3.3 \pm 1.8$ | $-90.9 \pm 2.1$ | $29.3 \pm 1.6$ | $-49.6 \pm 2.0$ | $71.7 \pm 7.0$ | $-28.4 \pm 5.8$ |
| ER-GS (Ours) | $2.9 \pm 1.4$ | $-90.6 \pm 1.1$ | $29.9 \pm 1.8$ | $-46.5 \pm 1.9$ | $\mathbf{79.7 \pm 4.7}$ | $\mathbf{-19.9 \pm 5.0}$ |
| ER-LS (Ours) | $4.5 \pm 1.2$ | $\underline{-86.7 \pm 1.5}$ | $\underline{33.4 \pm 1.0}$ | $\mathbf{-36.0 \pm 1.3}$ | $72.7 \pm 4.6$ | $-27.3 \pm 4.3$ |
| ER-GS-LS (Ours) | $\mathbf{39.5 \pm 3.1}$ | $\mathbf{-55.1 \pm 2.9}$ | $\mathbf{34.8 \pm 0.9}$ | $-23.9 \pm 1.1$ | $\underline{75.7 \pm 4.4}$ | $\underline{-24.2 \pm 5.1}$ |

Table 4: Performance comparisons on the task of Node-Unit Node Classification. **Bold**/underlined denote the best/second best-performing CL technique for each column. We note that non-CL baselines (first rows) serve as theoretical bounds for reference purposes. $-$ indicates ignored AF calculation.

| $K_n$ | CoraFull | | Arxiv | | Reddit | | Aromaticity | |
|---|---|---|---|---|---|---|---|---|
| | AP (%) ↑ | AF (%) ↑ | AP (%) ↑ | AF (%) ↑ | AP (%) ↑ | AF (%) ↑ | AP (%) ↑ | AF (%) ↑ |
| 0 | $2.9 \pm 1.4$ | $-90.6 \pm 1.1$ | $29.9 \pm 1.8$ | $-46.5 \pm 1.9$ | $79.7 \pm 4.7$ | $-19.9 \pm 5.0$ | $35.4 \pm 1.6$ | $6.2 \pm 0.7$ |
| 1 | $3.6 \pm 1.2$ | $-92.9 \pm 1.1$ | $31.5 \pm 1.2$ | $-37.6 \pm 1.3$ | $71.4 \pm 2.7$ | $-28.7 \pm 3.5$ | $33.5 \pm 1.2$ | $10.3 \pm 0.9$ |
| 3 | $4.8 \pm 1.1$ | $-91.8 \pm 0.9$ | $32.1 \pm 0.8$ | $-44.2 \pm 1.2$ | $73.8 \pm 2.6$ | $-26.1 \pm 2.9$ | $37.4 \pm 1.4$ | $13.5 \pm 0.8$ |
| 5 | $6.2 \pm 1.3$ | $-90.5 \pm 1.4$ | $33.9 \pm 0.5$ | $-38.2 \pm 0.4$ | $\mathbf{75.7 \pm 4.4}$ | $\mathbf{-24.2 \pm 5.1}$ | $42.8 \pm 1.2$ | $14.7 \pm 0.9$ |
| 10 | $26.6 \pm 2.7$ | $-67.8 \pm 2.3$ | $34.0 \pm 0.9$ | $-33.4 \pm 0.9$ | $72.1 \pm 3.6$ | $-27.9 \pm 4.1$ | $41.9 \pm 1.3$ | $14.6 \pm 1.5$ |
| 15 | $\mathbf{39.5 \pm 3.1}$ | $\mathbf{-55.1 \pm 2.9}$ | $\mathbf{34.8 \pm 0.9}$ | $-23.9 \pm 1.1$ | $73.7 \pm 3.5$ | $-26.3 \pm 3.9$ | $\mathbf{45.4 \pm 1.5}$ | $17.6 \pm 1.2$ |
| 20 | $38.2 \pm 3.2$ | $-56.28 \pm 2.8$ | $34.2 \pm 1.2$ | $-20.35 \pm 1.4$ | $72.1 \pm 3.8$ | $-27.1 \pm 4.2$ | $44.8 \pm 1.2$ | $\mathbf{10.7 \pm 0.9}$ |

Table 5: Performance comparisons on the selection of $K_n$ values for our ER-GS-LS. **Bold** denote the best-performing CL technique for each column.

be redundant or even harm global information. Therefore, the ER-GS-LS baseline performed worse than the Global Structure Distillation (ER-GS). Nevertheless, our method still has a gap of 70.6% AP and 70.3% AF with the Finetune baseline. With this result, we can confirm the usefulness of the Local-Global Knowledge Distillation in the node-level node classification task.

### 5.5 Model Analysis

In this section, we study the impact of the replay buffer on our distillation strategy. The hyperparameter $K_n$ determines the contribution of the samples from the replay buffer to the local and global distillation process. $K_n = 0$ is equivalent to ER-GS variant of our method. We examine different values of $K_n$ and report the performance of our method in Table 5. As the results demonstrate, sampling more nodes, i.e. larger $K_n$, in the graphs from the replay buffer brings significant improvement in both $AP$ and $AF$, especially in CoraFull, we have a substantial gap of 35% in $AP$ with $K_n = 15$. This result confirms the effectiveness of our method when the model was given samples in previous tasks to gain the learned local and global structure representation. In general, with a higher $K_n$ value, our model can be consistently improved; however, at $K_n = 20$, the performance stops improving, which might indicate we have reasonable local information to distil and increasing to a higher value might not help.

Finally, we examine our method with more advanced backbones besides GCN. The results consistently demonstrate that our proposed method is also compatible with GAT (Velivkovic et al., 2017) and GIN (Xu et al., 2018) backbones, improving them in continual learning tasks (see Appendix Table 7).

## 6 Discussion

In this paper, we systematically formulated Universal Graph Continual Learning (UGCL) and proposed a novel approach to prevent catastrophic forgetting issues under both node classification and graph classification settings. We extensively evaluated several common continual learning approaches on UGCL. The experiments show that standard regularization-based methods cannot effectively maintain prior knowledge in graphs, while rehearsal-based and parameter-isolation-based methods can perform better. The results also highlight

that our proposed solution using memory replay and subgraph representation constraints often achieves the best results. The method almost reaches the upper-bound result on both average accuracy and forgetting measure metrics. Our work is the first step toward the UGCL setting. In the following work, we will explore our method for large-scale graph problems such as scene graphs and transport networks.

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

## Appendix

| Dataset | $N_{\text{graphs}}$ | $N_{\text{classes}}$ | $N_{\text{tasks}}$ | $N_{\text{classes}}/N_{\text{tasks}}$ |
|---|---|---|---|---|
| Graph-unit node classification | | | | |
| SROIE | 973 | 4 | 4 | 1* |
| CORD | 1000 | 30 | 5 | 6 |
| WILDRECEIPT | 1739 | 12 | 4 | 3 |
| Graph-unit graph classification | | | | |
| ENZYMES | 600 | 6 | 3 | 2 |
| Aromaticity | 3868 | 30 | 15 | 2 |
| Node-unit node classification | | | | |
| CoraFull | 1 | 70 | 35 | 2 |
| Reddit | 1 | 40 | 20 | 2 |
| Arxiv | 1 | 40 | 20 | 2 |

Table 6: Description of benchmark datasets. *SROIE has #classes/task = 1 but is still valid because we provide an additional class named None/other in negative sampling.

| Method | GCN | | GAT | | GIN | |
|---|---|---|---|---|---|---|
| | AP (%) ↑ | AF (%) ↑ | AP (%) ↑ | AF (%) ↑ | AP (%) ↑ | AF (%) ↑ |
| Finetune | $4.9 \pm 0.0$ | $-87.0 \pm 1.5$ | $4.8 \pm 0.2$ | $-90.9 \pm 0.9$ | $4.8 \pm 0.1$ | $-88.2 \pm 0.8$ |
| Joint | $40.7 \pm 2.3$ | - | $43.4 \pm 0.8$ | - | $49.5 \pm 0.5$ | - |
| EWC | $4.9 \pm 0.0$ | $-88.9 \pm 0.3$ | $4.8 \pm 0.1$ | $-32.2 \pm 1.2$ | $4.9 \pm 0.1$ | $-90.3 \pm 0.6$ |
| MAS | $4.8 \pm 1.1$ | $-82.8 \pm 2.4$ | $6.5 \pm 0.5$ | $47.9 \pm 1.3$ | $4.7 \pm 0.1$ | $-84.6 \pm 0.8$ |
| GEM | $4.9 \pm 1.2$ | $-86.4 \pm 2.4$ | $4.9 \pm 0.3$ | $-88.6 \pm 2.2$ | $4.9 \pm 0.1$ | $-88.9 \pm 1.0$ |
| TWP | $4.8 \pm 0.9$ | $-88.3 \pm 1.5$ | $5.9 \pm 0.7$ | $-85.4 \pm 1.4$ | $4.9 \pm 0.1$ | $-89.9 \pm 0.2$ |
| LwF | $4.9 \pm 1.1$ | $-87.0 \pm 2.3$ | $4.9 \pm 0.6$ | $-89.4 \pm 2.4$ | $5.0 \pm 0.7$ | $-87.5 \pm 2.5$ |
| ER-GNN | $26.8 \pm 1.4$ | $-45.4 \pm 2.1$ | $\mathbf{26.7 \pm 1.6}$ | $-45.5 \pm 1.5$ | $30.6 \pm 0.3$ | $-52.5 \pm 0.4$ |
| ER (Ours) | $29.3 \pm 1.6$ | $-49.6 \pm 2.0$ | $\underline{23.3 \pm 1.2}$ | $-53.8 \pm 1.8$ | $\underline{31.4 \pm 1.2}$ | $-55.5 \pm 1.5$ |
| ER-GS (Ours) | $29.9 \pm 1.8$ | $-46.5 \pm 1.9$ | $14.3 \pm 1.3$ | $-65.3 \pm 1.7$ | $27.8 \pm 0.6$ | $-54.9 \pm 0.7$ |
| ER-LS (Ours) | $\underline{33.4 \pm 1.0}$ | $\underline{-36.0 \pm 1.3}$ | $24.7 \pm 0.8$ | $\mathbf{-30.9 \pm 2.8}$ | $29.4 \pm 1.1$ | $\underline{-49.1 \pm 0.9}$ |
| ER-GS-LS (Ours) | $\mathbf{34.8 \pm 0.9}$ | $\mathbf{-23.9 \pm 1.1}$ | $21.3 \pm 2.3$ | $\underline{-31.5 \pm 1.4}$ | $\mathbf{32.6 \pm 1.2}$ | $\mathbf{-44.5 \pm 0.1}$ |

Table 7: Performance comparisons on the task of Node-Unit Node Classification with 3 different backbones on Arxiv-CL dataset. **Bold**/underlined denote the best/second best-performing CL technique for each column. We note that non-CL baselines (first rows) serve as theoretical bounds for reference purposes. $-$ indicates ignored AF calculation.

| Number of replay nodes per class | CoraFull | | Arxiv | | Reddit | |
|---|---|---|---|---|---|---|
| | AP (%) ↑ | AF (%) ↑ | AP (%) ↑ | AF (%) ↑ | AP (%) ↑ | AF (%) ↑ |
| 10 | $\mathbf{2.1 \pm 1.1}$ | $-91.6 \pm 1.7$ | $\mathbf{26.8 \pm 1.4}$ | $-52.4 \pm 2.1$ | $\underline{36.2 \pm 2.2}$ | $\underline{-66.0 \pm 2.5}$ |
| 100 | $2.0 \pm 0.5$ | $-92.5 \pm 1.3$ | $\underline{25.8 \pm 1.2}$ | $\underline{-59.6 \pm 2.3}$ | $\mathbf{71.5 \pm 5.2}$ | $\mathbf{-28.6 \pm 2.4}$ |
| 1000 | $2.0 \pm 1.2$ | $-92.0 \pm 1.6$ | $5.8 \pm 1.2$ | $-87.5 \pm 2.5$ | $23.2 \pm 1.2$ | $-79.5 \pm 2.5$ |

Table 8: Performance comparisons of ERGNN (Zhou & Cao, 2021) on the task of Node-Unit Node Classification with 3 different replay buffer sizes. We report the number of replay nodes per class because datasets have different numbers of nodes and thus need to scale differently. **Bold**/underlined denote the best/second best-performing CL technique for each column. We note that non-CL baselines (first rows) serve as theoretical bounds for reference purposes. $-$ indicates ignored AF calculation.

