# OpenReview forum: "Universal Graph Continual Learning"
_TMLR — Accepted by TMLR_

### Review · Reviewer_sgKy · 2023-09-18

**Summary Of Contributions:**

This paper introduces the concept of Universal Graph Continual Learning (UGCL) and presents a novel approach to mitigating catastrophic forgetting in both node and graph classification scenarios. Proposed methods are evaluated against several conventional continual learning approaches applied to UGCL. Results consistently demonstrate that the solution, incorporating memory replay and subgraph representation constraints, frequently outperforms other methods.

**Audience:**

Yes

**Broader Impact Concerns:**

Nothing.

**Claims And Evidence:**

Yes

**Requested Changes:**

1. To underscore the practical importance of the three problems (GUGC, GNUC, NUNC), authors should include real-world examples illustrating the significance of the study
2. How to address the disparity in node and graph sizes requires further explanation regarding how this difference impacts the buffer.
3. It is crucial to compare the proposed approach with existing Graph Continual Learning methods. This is particularly important as most current baselines in the paper are not specifically tailored for Graph Neural Networks (GNNs)."
4. The notation should be more friendly. For instance, θ in Equations 2 and 3 is typically used to denote a parameter. Additionally, the alphas in Equation 1 and Equation 6 should be guaranteed to be consistent.
5. Are there plans to make the Benchmark publicly accessible?

**Strengths And Weaknesses:**

Strengths:
This work introduces a comprehensive taxonomy for graph continual learning and presents a novel method designed to address all categories within this taxonomy. Additionally, it establishes a new benchmark for continual learning in graph-based node classification, following a class-incremental setting.

Weaknesses:
1. To underscore the practical importance of the three problems (GUGC, GNUC, NUNC), authors should include real-world examples illustrating the significance of the study
2. How to address the disparity in node and graph sizes requires further explanation regarding how this difference impacts the buffer.
3. It is crucial to compare the proposed approach with existing Graph Continual Learning methods. This is particularly important as most current baselines in the paper are not specifically tailored for Graph Neural Networks (GNNs)."
4. The notation should be more friendly. For instance, θ in Equations 2 and 3 is typically used to denote a parameter. Additionally, the alphas in Equation 1 and Equation 6 should be guaranteed to be consistent.

---

> ### Author Response · Authors · 2023-10-16
> **Reply to Reviewer sgKy**
>
> Thank you for your valuable feedback. We address your questions and concerns below.
> 1. We have revised the introduction to include real-world examples to highlight the significance of our study. Thank you for your suggestion.
> 2. A solution for the disparity problem is to finetune the number of replay samples (nodes/ graphs) separately for each dataset. We have added a new experiment to illustrate our point. According to Appendix Table 8 in this revision of our paper, datasets with few nodes like Corafull are less affected by the number of replay nodes. As the number of nodes increases, the gap between using 10 and 100 replay nodes per class becomes bigger (Arxiv then Reddit). Interestingly, none of them benefit from using 1000 replay nodes per class. and some even worse like Reddit. Using too many replay nodes could make the model focus too much on past data thus less focus on the current task.  In the real-world scenario when the number of incoming data is unknown, we can pre-allocate a very big replay buffer with more than enough capacity. After that, try to do hyperparameter tuning of the number of actual samples that need to be stored on the current data stream (e.g: use only 100 samples per class at most instead of the maximum capacity of 1000 samples). In the end, the model stability is the signal for the right tuning.
> 3. Thank you for your suggestion. We have added a new baseline ER-GNN that is designed for CL in graph neural networks. The results show that our method overall outperforms ER-GNN significantly.
> 4. We have fixed our notation as recommended. We also updated Eq. 1 and 6 to clarify the usage of alpha.
> 5. Yes, we will release our benchmark code after our paper is published.

---

### Review · Reviewer_8eiQ · 2023-09-28

**Summary Of Contributions:**

The paper is about Universal Graph Continual Learning, which is an interesting topic. The authors propose a novel method that enables graph neural networks to excel in this universal setting. The paper is well written and well organized. However, there are several concerns in the current version of the paper that addressing them will increase the quality of this paper.

**Audience:**

Yes

**Claims And Evidence:**

Yes

**Requested Changes:**

1 Improved summary writing.
2 Adjust Figure 1.
3 Add more experiments.

**Strengths And Weaknesses:**

Strengths

>1 The topic the paper focuses on is meaningful, and the authors try to address the deep problems in the field.

>2 The paper is well structured and easy to understand.

>3 The authors conducted experiments to support their ideas.

Weakness

>1 The abstract section does not provide sufficient background and further additions are recommended.

>2 In the loss function, there are three additional constraint terms in addition to the main task loss. Although these constraint terms are tuned by parameters, I still hold the doubt whether all of these constraint terms really play a decisive role.

>3 The font size of Figure 1 is small and the information conveyed is limited; it is recommended that the figure be further improved to fully demonstrate the modeling structure of the paper.

>4 I hold a confusion, according to the author, in the past, the task class data information to be stored in the buffer, so if the amount of data accumulated more and more, whether the memory occupation will be a problem?

>5 Experimentally, did the authors use a smaller dataset? The methods compared are more classical algorithms, but are there emerging methods that have not been compared?

---

> ### Author Response · Authors · 2023-10-16
> **Reply to Reviewer 8eiQ**
>
> Thank you for your constructive review. We address your questions and concerns below.
> 1. We have revised the abstract as recommended.
> 2. In the original version of our paper, we already reported the results when we removed the terms incrementally in each result table to prove that all of the terms are crucial for the best performance. Here we explain again the meaning of our ablated models:
>    - Finetune: only task loss is used, i.e., none of the three terms are used.
>    - ER: only experience replay loss is used, i.e.,  removing global and local structure losses $\beta=\gamma=0$
>    - ER-GS: experience replay and global structure losses are used, i.e, removing local structure loss $\beta=0$
>    - ER-LS: experience replay and local structure losses are used, i.e, removing global structure loss $\gamma=0$
>    - ER-LS-GS: all three losses are used
> Overall, the results show that ablating any of our loss terms will reduce the continual learning performance. We hope that our answer has cleared up your doubts.
> 3. We have improved the Figure as suggested
> 4. Thank you for your interesting question. When the amount of data accumulates more and more, the memory occupation will be a problem indeed. That is the reason why we introduce a new version of the experience replay compared to the well-known ERGNN. In our new replay mechanism, we introduce a replacement strategy when the memory is full. We use a class-balancing replacement strategy which is optimal for maintaining the accuracy of all classes equally
> 5. Our set of data includes 8 different datasets, whose sizes vary from small to big.  For example, ENZYMES is a small dataset of 600 data points. We believe our datasets are comprehensive and well-representing practical graph continual learning problems. Although most of our baselines are classic, they are common models used in benchmarking graph CL (Zhang et al., 2022). In the new version, we have added a more recent baseline ER-GNN which also utilizes replay mechanisms to improve graph CL. Our method overall outperforms ER-GNN significantly.

---

### Review · Reviewer_STKY · 2023-10-02

**Summary Of Contributions:**

The authors propose a new method for graph continual learning, that is based on mixing several techniques for this task. The method composed of three main components: Experience Replay (ER),Global Knowledge distillation (GS), and  Local Knowledge Distillation (LS).

The authors provide many experiments to show the performance of their method, that often times improves existing methods.

**Audience:**

Yes

**Claims And Evidence:**

Yes

**Requested Changes:**

Please see questions in my review.

**Strengths And Weaknesses:**

Strengths:
- I think that the paper is nicely written, and gives a very good explanation of the problem to be solved, existing solution, and the proposed method itself is very clear. Also, I like it that the authors discuss each method compared with in the experimental section.

-The results seem promising and offer a universal approach, as suggested by the authors.

Weaknesses/Questions:
- How would your performance change if you use another GNN backbone instead of GCN?
- How did you choose to use 3 layers? What is the performance when using more or less layers?
- In some cases your method does very well, for example in CORD dataset, but less so in other datasets. Can you please explain why?
- I understand that your loss contains several hyper parameters, which is fair. However I think that it would be nice if you can provide some intuition about their weighting and an ablation study of their influence on the results.

---

> ### Author Response · Authors · 2023-10-16
> **Reply to Reviewer STKY**
>
> Thank you for your thoughtful comments. We address your questions and concerns below.
> - We have added experiments using different backbones (GAT and GIN) instead of GCN. As shown in Appendix Table 7, our method still works well with other backbones.
> - We would like to clarify that we used 3-layer GCN for the document understanding task, which was a common architecture used for datasets like CORD. For other tasks, we adopted the default network architecture from the CGLB benchmark (Zhang et al., 2022), which can use a different number of layers depending on the backbone (GCN, GAT, GIN). We have done a quick test and realized that 3 or fewer layer-GCN do not show much difference in terms of performance. More layers can destroy the performance of all CL methods.
> - Overall, our proposed method outperforms all other baselines by a large margin. In some datasets, the gap is bigger than the other datasets in the same domain because of the task difficulty difference. The task difficulty is defined by the number of classes per task as well as the number of tasks. As a rule of thumb, the longer the sequence, the more difficult it is to prevent catastrophic forgetting. An example of this point is Aromaticity versus Enzymes where the gap between our method and the plain experience replay is bigger for Aromaticity. Then, the number of classes per task also makes it more difficult to keep the accuracy drop less. For example, in CORD 6 classes per task while in SROIE we have only 1. Thus, the gap in CORD is bigger than in SROIE
> - In the original version of our paper, we already reported the ablation study in each result table to prove that all of the losses are crucial for the best performance. Here we explain again the meaning of our ablated models:
>    - Finetune: only task loss is used, i.e., none of the three terms are used.
>    - ER: only experience replay loss is used, i.e.,  removing global and local structure losses $\beta=\gamma=0$
>    - ER-GS: experience replay and global structure losses are used, i.e, removing local structure loss $\beta=0$
>    - ER-LS: experience replay and local structure losses are used, i.e, removing global structure loss $\gamma=0$
>    - ER-LS-GS: all three losses are used
>
> Overall, the results show that ablating any of our losses will reduce the continual learning performance. We have tuned $\beta$ and $\gamma$ and realized that a balance contribution of local and global losses got the best result ($\beta=\gamma=0.5$). We have revised our paper (Eq. 1 and 6) to clarify the usage of $\alpha$ and in practice, we do not need to tune $\alpha$.

---

### Author Response · Authors · 2023-10-15
**General response and summary of changes**

Dear Reviewers,

We thank the Reviewers for your review effort and valuable feedback. We appreciate positive comments and have improved our paper to address your concerns. We will reply to each Reviewer to answer your specific questions. Here, we summarize the changes in this revision below,
- Improve the writing of the abstract and introduction
- Improve the clarity of our Figures and equations
- Add a new graph continual learning baseline ER-GNN (Zhang et al., 2022)
- Add new experiments with different graph neural network backbones

---

### Author Response · Authors · 2023-12-04
**Github link**

Please refer to our implementation in this link: https://github.com/doviettung96/CGLB
We implement our method from the original work of CGLB (cited in our paper).

---

> ### Public Comment · ~Lei_Song1 · 2024-11-15
> **Where is the implementation of ER-GS-LS?**
>
> The link you provided doesn’t seem to contain the implementation of ER-GS-LS.

---

> > ### Author Response · Authors · 2024-11-22
> > **We are working on it**
> >
> > Hi, sorry that the code link provided no longer contains ER-GS-LS. We are creating a new repository to host the paper's code.

---

### Decision · Action_Editor_dsau · 2023-11-06

**Recommendation:** Accept as is

**Comment:**

Overall the reviewers agree that the topic is relevant and meaningful to the TMLR community and that the contribution is technically correct. The discussion period allowed the authors to improve the paper by incorporating feedback from the reviewers.

**Audience:**

Yes, the problems of continual learning and learning on graphs are very relevant to TMLR's audience.

**Claims And Evidence:**

Yes. Some minor concerns were initially raised by some of the reviewers about how convincing the experiments were but they were clarified during discussion and in the revision.